

**Local perception and adaptation strategies to landslide occurrence in the Kivu catchment of Rwanda**

Ma-Lyse Nema[1], Prof. Bachir Saley Mahaman [1], Dr. Arona Diedhiou[1], Assiel Mugabe[2]
[1]: University of Felix Houphouet Boigny, Ivory Coast
[2]: Rwanda Polytechnic- IPRC Kigali, Rwanda
**Correspondence address:** nemalyse@gmail.com  Tel: +2250545426006/+250787042233

**Abstract**

This study aims to assess local perception and adaptation strategies to landslide occurrences in the Kivu
catchment of Rwanda. Qualitative, quantitative, and combined methods were applied for data collection to
investigate how the locals perceived the types of landslides, origins, impacts, contributing variables to
landslides, and adaptation strategies to landslides. The investigation conducted interviews with 384 residents
from the six districts of the study area, key informant interviews and field observations. The findings showed
that falls, flows, slides and spreads are the types of landslides frequently occurring in the study area. Heavy rain,
steep slope, road construction, inappropriate agriculture practices, deforestation, earthquake, and mining were
found to cause landslides with effects such as human fatalities, infrastructure damage, injuries, and property
losses. Different measures are adopted for landslide risk reduction, including agroforestry, terracing, stormwater
drainage systems, and relocating people from high-risk areas. Residents have a positive opinion of their
community's approach to managing landslides effectively. However, the findings revealed gaps in cooperation
between the parties where Non-Governmental Organizations do not appear to be active participants during
intervention activities for landslide management in the study area. Regression analysis has shown that
deforestation, inappropriate agricultural practices, roads construction, earthquakes, and climate change are the
key factors significantly contributing to landslide occurrences in the study area. Further research must be
conducted on the subject using a variety of methodologies, notably those related to applied artificial intelligence
in order to enrich the literature presently available on landslides in the Kivu catchment of Rwanda.

**Keywords**: Landslides, local perception, Kivu catchment of Rwanda, adaptation strategies.

## 1. Introduction

Landslides are included among big natural hazards which cause the loss of lives and property damages in
different regions of the world (Fu et al., n.d.; Peng et al., 2022). Therefore, there are many fatalities and losses
encountered all over the world; and research discoveries showed them to be engendered by landslide
occurrences. The findings showed also that hilly regions around the world are the most threatened by
landslides(Jamal & Abdolkhani 2009; Odhiambo et al., 2019). Based on their movement mechanism and the
elements that compose the slope, landslides can be divided into distinct types namely falls, topples, spreads,
flows, and slides, and each category is influenced by several factors, including topographic features, slope-
forming materials, and external impacts(Highland, 2004b). Many years ago, and even in recent times,
researchers have developed theories and assumptions about the factors fueling landslide occurrences. The main
causes of landslides were found to be a result of complex combination of physical factors, geological factors,





morphological factors, anthropogenic factors, climatological factors, and hydrological factors (Claude et al., 2020; Walker and Shiels, 2013).

This complexity holds also other important factors leading to landslides occurrences; and research discoveries mentioned them as land use, slope angle, slope length, internal relief, and other many more ( Nsengiyumva et al., 2018a.; Nsengiyumva et al., 2018b; Nsengiyumva & Valentino, 2020) . Landslides can also be brought on

by certain human activity, such as mining, clearing slopes for buildings and roads, and deforestation (Maki Mateso et al., n.d.; Senouci, 2020). Other scientists particularly cite weathering of materials, head stress, higher slope angles, changes in groundwater level, and eradication of flora as additional reasons of landslides (Brönnimann, 2011.; Igwe, 2015; Jeong et al., 2017).

On the other hand, people's perceptions on landslide factors can differ based on their experience and level of

familiarity with landslides. Capacity reflects the community's ability to mitigate the risk of landslides. This is relevant to several aspects, one of which is the community's perception and knowledge of landslides. Knowing how people see things will help the community to respond to and deal with predicted future tragedies (Setiawan et al., 2014). The way society is perceived, is influenced by several intricate factors including personal, social, cultural psychological, economic, and political ones(Wisner & Nivaran, 2003).

The distribution of landslide disasters around the world is not uniform because it relies on the geology, population, quantity of rainfall, and regional environment (Nakileza et al., 2017). On a global scale, a few regions have been recognized as prone to landslides. There are five regions in Asia that have been recognized as being prone to landslides, including the southern edge of the Himalaya, Taiwan, Indonesia, and Central China(Forbes et al., 2012a). The Pacific Coastal Ranges, Rocky Mountains and Appalachian Mountains in the

United States are well recognized for their susceptibility to landslides (Mirus et al., 2020).

Landslides regularly happen in unplanned communities, cities, and rural areas in the Eastern Caribbean that are situated on steep terrain (Anderson et al., 2011). Numerous sites in Africa have been classified as landslide-prone areas. Equatorial Africa, East Africa, particularly the Rwenzori Mountains and Congo Nile Crest, the Cameroon Volcanic Line, Kivu region of the Democratic Republic of the Congo and the Elgon areas of Uganda

are among them (Kervyn et al., 2018).

According to Birkmann & Wisner (1972) more than 3 billion people have been impacted by disasters over the years. Given that the entire effects of natural catastrophes are not often reported, the statistics showed that more than 750,000 persons died, and 600 billion dollars' worth of property damage was recorded. Because of the number of fatalities and property damage they produce, landslides are ranked as the third most deadly disaster in

the world (Nsengiyumva et al., 2018b) . According to Anderson et al., (2011), between 2004 and 2010, landslides claimed the lives of almost 30,000 people globally. In the USA alone, landslides claim the lives of between 25 and 50 people per year. In addition, landslides claimed the lives of 4,369 individuals worldwide in 2015, affecting 150,332 people directly or indirectly. The death toll rose to 32,322 in 2016 (McAdoo et al., 2018). In Sri Lanka, where landslides are common, 50,000 people have been affected in the past five years

(McAdoo et al., 2018). Statistics indicate that during the 1970s, landslides have grown more than five times in Asia. There were 88 landslides recorded between 2000 and 2009, and 5,367 persons died as a result (Forbes et al., 2012b).



With careful planning and control, landslide-related social and financial losses can be minimized. These strategies include (a) restricting development in landslide-prone locations, (b) using excavation, grading, landscaping, and construction codes, (c), using physical measures (drainage, slope-geometry alteration, and structures), and (d) developing warning systems (Maes et al., 2017; Schuster& Kockelman,1982). These techniques, according to Schuster & Highland (2007), could minimize landslide losses in California by more than 90%. According to Dai et al., (2001) , implementation of these strategies had already resulted in a 92–97% decrease in landslide losses in the City of Los Angeles. Nevertheless, landslide activity is rising globally despite advancements in hazard assessment, prediction, mitigation, measures, and warning systems.

Measures to reduce the danger of landslide to the community may be implemented, if necessary, once the risk from a landslide or areas susceptible to or affected by landsliding is identified. Planning control, which can be performed by removing or converting existing development, discouraging, or restricting new development in unstable places, is one way that the community dealing with landslide can cope with (Schuster and Fleming, 1986; Turner & Schuster, 1996; Kockelman, 1986).

All over the world, governments play a crucial role in landslide risk reduction through institutions and agencies tasked with reducing the risk of landslide tragedy (Royal Institution of Chartered Surveyors., 2010).Thus, preventing and responding to landslides involves many institutions and sectors. Most governments have established national disaster response agencies, which may be decentralized to the regional, district, and village levels. The government is in charge of community development and long-term landslide risk reduction, and it is also the first response (ISDR, 2010). In order to ensure the successful implementation of policies meant to reduce the risk of landslides, the government further empowers local governments as one of its top priorities(Royal Institution of Chartered Surveyors., 2010).

The community's contribution to disaster management may be observed in their participation in all aspects of disaster management as well as in the local development of landslide risk reduction policies, strategies, and plans. The involvement of communities in rescue and recovery efforts, as well as in fundraising efforts to aid victims and rehabilitation efforts, was highlighted by Birkmann & Wisner (2006b).

In the past, landslides have had an impact on many parts of Rwanda. Landslides have caused fatalities, serious injuries, and many people to lose their homes and properties. The statistics show that between July 2016 and June 2018, landslides caused 93 deaths, 39 injuries, 1362 damaged houses, 802.98ha of crops destroyed and 128 livestock lost. Furthermore, 20 roads and 25 bridges were damaged in different parts of Rwanda (MINEMA,2018). Moreover, the records show that between January and December 2020 landslide caused 125 deaths, 63 injuries, 1643 damaged houses, 180 ha of crops destroyed, 170 livestock lost, 71 roads damaged, 43 bridges destroyed, 4 classrooms destroyed, 1 health center destroyed, 16 water supply and 3 electricity transmission lines damaged in different areas of Rwanda (MINEMA,2020).

However, there hasn't been much research and literature regarding the risks of landslides in Rwanda. Additionally, there are large data gaps on previous landslide occurrences (MIDIMAR, 2015). Rwanda Ministry of Disaster Management and Refugee Affairs (MIDIMAR) began systematically cataloging disasters in 2010. Before this time, the only sources of disaster data were worldwide data gathering centers like the Royal Museum for Central Africa (RMCA) and CRED (EM-DAT). The recorded occurrences are typically not accurately



georeferenced, and the inventory is difficult. It was remarked that available reports do not separate landslide with flooding, which makes it difficult to acquire specific information on landslide risk management in the study area. Furthermore, the literature showed gaps in data and few research for landslide risk management in the Kivu catchment of Rwanda. Therefore, these gaps should be a challenge not only to the researchers but also to different partners in landside risk management in Rwanda.

## 2. Materials and methods

### 2.1. Description of the study area

Extended to 2425km², the Kivu catchment of Rwanda is found between 28°55′30″S and 29°26′00″S latitude, 10°33′E and 20°33′E longitude, eastwards to the surroundings of Lake Kivu. The zone under study is landlocked with six districts specifically Rusizi, Nyamasheke, Karongi, Rutsiro, Nyabihu and Rubavu. Its altitude ranges from 1700 meters to 4507 meters (Byers, 1992; Mikova et al., 2015). The rainfall ranges between 1000 to 1200mm/year but sometimes it increases up to 1800mm/year and seasonal year is partitioned into rainy and dry season (Basnet and Vodacek, 2015).

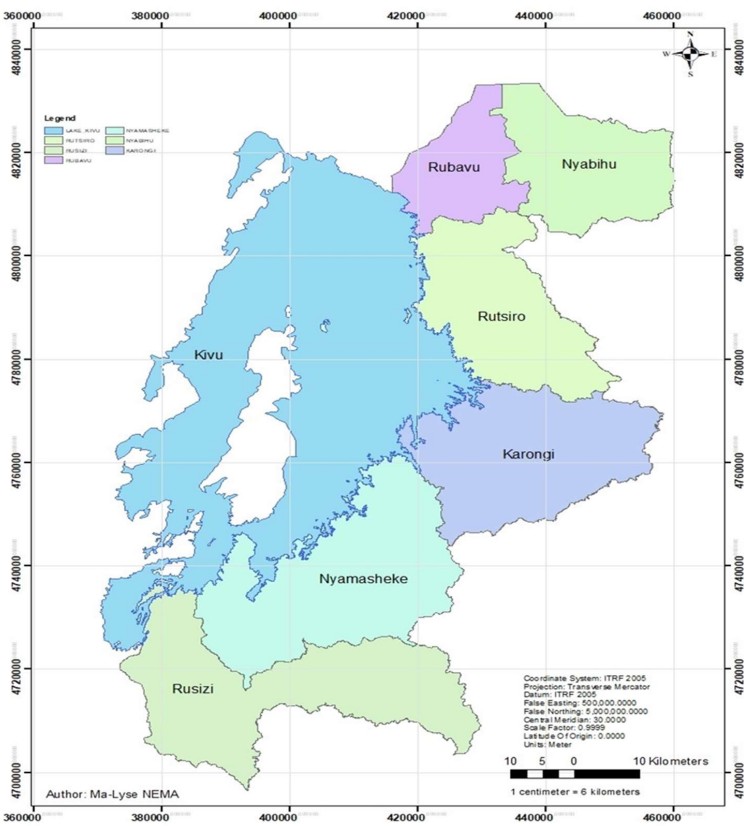

*Figure 1: The Kivu catchment of Rwanda (study area).*



### 2.2. Data collection

The data in this study was collected for a period of three months. The sample size for the general population, as suggested by (Sekaran, 2003), was set at 384 residents based on the estimated population of the study region of 2,142,096. The local residents interviewed were selected from different cells. The key informants were questioned about the causes of landslides in the region, their role in reducing landslide risk, their collaboration with neighborhood non-profit organizations in landslide management, the difficulties encountered in landslide management, and future plans for landslide risk reduction.

The study used a variety of sampling methods. Landslide sites were chosen using simple random sampling, while people at particular places were chosen using systematic random sampling. Since it lessens the impact of auxiliary factors, this type of randomization increases the validity and dependability of the data (Oso and Onen,2008). The same method was used in selecting key informants including local leaders and experts from Districts, Ministry of emergency management and University.

Key informant (KI) interviews, observations, and interviews with local inhabitants were among the qualitative methods utilized to obtain the data. By distributing questionnaires directly to the chosen residents and making notes during key informant interviews and observations, qualitative data was gathered. With the use of a checklist, an observation checklist was created, and systematic observations were performed at the locations where landslides had occurred. These observations centered on the locations of the landslide-prone areas, the variables that make them vulnerable to landslides, the usage of the land, human activity, the resilience of landslides, and nearby mitigation-related actions. ODK was used to digitize the surveys, and tablets with KoboCollect v2922.2.3 loaded were used to collect the data.

According to the criteria provided by the researcher, the data was cleaned and coded. Data was then collected after the coding stage and uploaded to SPSS Software for statistical analysis. Research assistants were hired and given a day of training on how to conduct the questionnaires and gather high-quality qualitative data to ensure quality. They are fluent in local language (Kinyarwanda) and English. They were also familiar with the study region and local culture. The pre-tested questionnaires were revised as needed to fill in any gaps found after pre-testing. Daily meetings with research assistants were also arranged to address any issues that came up during the data collection procedure. The researcher was present in the field and regularly supervised the data collection exercise.

Besides quality assurance, ethics were considered in data collection. The respondents were verbally informed of the study's goal during the data collection process, and their consent to participate was gained. The respondents were made aware that their participation was completely optional, private, and risk-free, and that they might decide not to participate at any point throughout the interview.

### 3. Results and discussions

The number of respondents in this survey for Kivu catchment of Rwanda were 384 including 51.8% males and 48.2% females. The categories of respondent's age were found to be below 25, 25-40, 40-55, and above 55; and the percentages for each category were 15.4%, 39.1%, 36.5% and 9.1% respectively. The marital status of the respondents showed the categories of divorced, married, separated and single with 1%, 70.6%, 5.7% and 22.7%





respectively. The percentage breakdown of the completion of primary school, secondary school and university
levels were 21.9%, 55.5% and 19.3 % respectively. The remaining 3.4% of respondents were found in the
category of illiterate. Looking at education fields, the survey showed that 9.9% have background in agriculture
and environment and 31% have schooled in social and health sciences. It was found that 21.9% of respondents
have educational background in other domains such as technical and vocational education, literature, and others.
However, 37.2% of the respondents were found not applicable in the domains mentioned. The duration of
residing in the area was below 5 years, 5-10 years, 10-20 years and above 20 years, rating 4.7%, 34.1%, 34.1%
and 27.1% respectively.

*Table 1: Social Demographic characteristics of respondents*

| District | | Karongi | Nyabihu | Nyamasheke | Rubavu | Rusizi | Rutsiro | Total | Percentage |
|---|---|---|---|---|---|---|---|---|---|
| Sample | | 71 | 31 | 82 | 32 | 52 | 116 | 384 | |
| Gender | Female | 33 | 12 | 40 | 16 | 26 | 58 | 185 | 48.2% |
| | Male | 38 | 19 | 42 | 16 | 26 | 58 | 199 | 51.8% |
| Marital status | Divorced | 0 | 0 | 1 | 1 | 2 | 0 | 4 | 1.0% |
| | Married | 41 | 17 | 61 | 27 | 37 | 88 | 271 | 70.6% |
| | Separated | 1 | 1 | 5 | 2 | 0 | 13 | 22 | 5.7% |
| | Single | 29 | 13 | 15 | 2 | 13 | 15 | 87 | 22.7% |
| Age range | Below 25 years | 21 | 11 | 9 | | 10 | 8 | 59 | 15.4% |
| | 25-40 | 26 | 5 | 35 | 14 | 24 | 46 | 150 | 39.1% |
| | 40-55 | 18 | 9 | 31 | 15 | 15 | 52 | 140 | 36.5% |
| | Above 55 | 6 | 6 | 7 | 3 | 3 | 10 | 35 | 9.1% |
| Education level | None | 2 | | 2 | | 1 | 8 | 13 | 3.4% |
| | Primary | 17 | 10 | 16 | 6 | 14 | 21 | 84 | 21.9% |
| | Secondary | 44 | 19 | 41 | 19 | 28 | 62 | 213 | 55.5% |
| | University | 8 | 2 | 23 | 7 | 9 | 25 | 74 | 19.3% |
| Field of education | Agriculture and environment | 4 | 4 | 16 | 1 | 6 | 7 | 38 | 9.9% |
| | Social and health sciences | 14 | 3 | 27 | 10 | 16 | 49 | 119 | 31.0% |
| | Other | 21 | 6 | 19 | 8 | 10 | 20 | 84 | 21.9% |
| | Not applied | 32 | 18 | 20 | 13 | 20 | 40 | 143 | 37.2% |

Source: Field survey, Kivu catchment of Rwanda, 2022

The factors such as heavy rain, steep slope, road construction, inappropriate agriculture practices, deforestation,
earthquake, and mining were attributed by 37%, 29%, 11%,11%,9%,2%, 1% of respondents to be the cause of
landslides in Kivu catchment of Rwanda. Both heavy rain and topography (steep slope) were ranked to be the
major causal factors for landslides occurrences. According to Rahman (2017) and Akhter (2017), Hilly soils
become soft and loose and slide down to the surface when heavy rainfall seeps inside through the soil and
infiltrates beneath the surface, applying pressure to the subsoil. The Kivu catchment of Rwanda is among hilly
areas of the country with altitude ranging between 1700 meters and 4507 meters (Byers, 1992; Mikova et al.,
2015). It is also known for its high rainfall which can reach 1800mm per year. The results from the survey on
causal factors for landslide occurrences in the study area are in conformity with previous research on natural
disasters occurred in Western Province of Rwanda. On 13 May 2021, Richard Davies, the author in
floodlist.com reported landslide occurrence cases after a period of intense rain on May 9, 2021.  Consequently,
there were around 100 residences affected, 39 of which were apparently entirely demolished. According to
Western Province Government officials, this resulted in the displacement of 631 persons from 117 families.



Although heavy rainy was revealed to be the primary cause of landslide occurrences in Rwanda's western province, it is important to remember that rainfall is not the only one causal factor for landslide, so it is important to keep other factors in mind as well. Within this context, the study showed that earthquakes also affect landslides in the study area. The question remains as to why earthquakes were brought up in this specific conversation when the results only gave it a 2% rating! The answer to this question is justified by variances in the lithology and geological formation of the six districts of the Kivu watershed (Rusizi, Nyamasheke, Karongi, Rutsiro, Nyabihu, and Rubavu).

To get back on track, the District of Rubavu is in Northwestern region of Rwanda which experiences earthquakes. Indeed, as a single district within Kivu catchment of Rwanda that touches close to an active volcano of Nyiragongo may be justified to be exposed to landslides induced by earthquake. Furthermore, earthquakes in the Rubavu District undermine the fragile volcanic soils, causing landslides.

*Table 2. Triggering factors of landslide occurrences in the study area*

| Factors | Frequency | Percentage |
|---|---|---|
| Heavy rain | 142 | 37% |
| Steep slope | 111 | 29% |
| Road's construction | 42 | 11% |
| Inappropriate agriculture practices | 42 | 11% |
| Deforestation | 35 | 9% |
| Earthquakes | 8 | 2% |
| Mining | 4 | 1% |
| Total | 384 | 100% |

Source: Field study, September- October 2022 (Note: Multiple responses were considered)

Four different types of landslides namely falls flows, slides and spreads were found in Kivu catchment of Rwanda according to survey findings. The frequency of these landslides varies throughout the six districts (Rusizi, Nyamasheke, Karongi, Rutsiro, Nyabihu and Rubavu) that make up the catchment. Slide is the most common type of landslide identified in this survey, and it was evaluated at 73.4% across all districts. In the study region, it was discovered that the Rutsiro district had the most frequent sliding incidents. The three districts (Nyamasheke, Rusizi Karongi, and Rubavu) displayed almost similar rates in the occurrences of slides, although Nyabihu displayed the lowest rank. According to the survey, the majority of flow types were found in the districts of Nyamasheke and Karongi, while just a few occurrences were detected in the other districts of the research area.

Except for Rubavu district, falls were discovered in the five districts; and the district of Karongi was determined to have the most landslides in the form of falls. The physical features of the Karongi district (highlands with steep slopes) and the preponderance of bed rocks along steep slopes of roadways may be responsible for the district's high prevalence of falls.

*Table 3. Types of landslides per District*

| Districts | Falls | Flows | Slides | Spreads |
|---|---|---|---|---|
| Karongi | 12 | 23 | 33 | 3 |
| Nyabihu | 4 | 12 | 14 | 1 |




| Nyamasheke | 3 | 29 | 50 | 0 |
|---|---|---|---|---|
| Rubavu | 0 | 0 | 32 | 0 |
| Rusizi | 2 | 2 | 47 | 1 |
| Rutsiro | 7 | 3 | 106 | 0 |
| Total | 28 | 69 | 282 | 5 |
| Percentage | 7.3% | 18.0% | 73.4% | 1.3% |

Source: Field study, September- October 2022 (Note: Multiple responses were considered)

Few studies have tried to quantify how landslides affect socio-economic systems (Mertens et al., 2017). The survey conducted on local perception on landslide occurrences in Kivu catchment of Rwanda revealed the socio-economic impact of landslide in the study area.

In this context, the various effects were ranked as follows: 43% for human fatalities, 43% for infrastructure damage, 9% for injuries, and 5% for property losses.

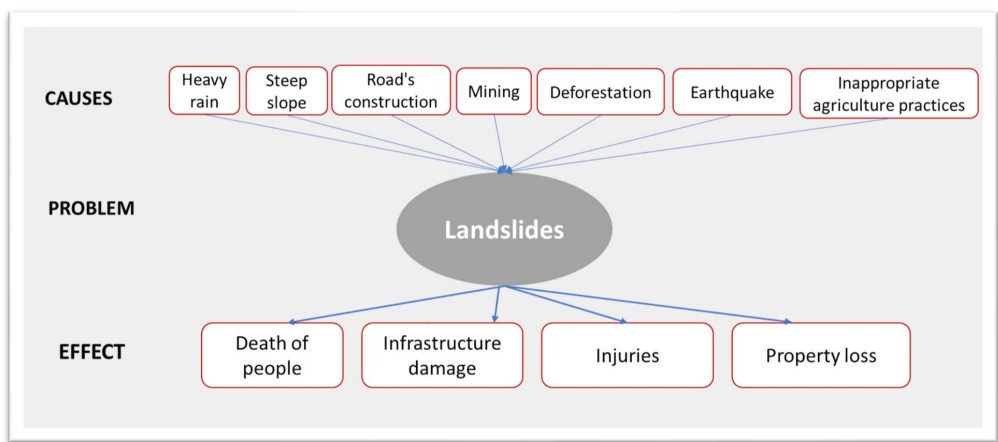

*Figure 2. Landslides Causes-effects diagram of Kivu catchment. (Source: Residents, field survey, September-October 2022).*

Different sectors are impacted both directly and indirectly in disasters of all kinds, landslides in particular. As a result, a connection is made between these sectors, factors that contribute to vulnerability, and related impacts to detail the actual situation for effective intervention efforts. Through its national contingency plan for floods and landslides of 2018, the Ministry of Emergency Management (MINEMA) clarified the impact of such disasters in many sectors across the entire nation of Rwanda. Since the Kivu catchment of Rwanda represents a region vulnerable to landslides and their impacts in the sectors described in this section, their findings ought to provide a good picture of the actual situation there.

A growing number of disaster threats, such as landslides and others, have been experienced in Rwanda; with severity and consequently disrupting people's livelihoods, causing significant financial losses, losses to the environment, interruptions to economic activity, and socioeconomic retardation. Different measures were taken for landslide reduction, according to a survey on current landslide control strategies in Kivu catchment of Rwanda. Agroforestry is one of these measures, according to 32% of respondents. 24 percent of the respondents considered relocating people from high-risk areas as important. Another measure for reducing landslides in the



research area is terracing, which was ranked by 22% of respondents. Additionally, 20% of respondents rated installing stormwater drainage systems as important, and 2% said that growing trees along riverbanks already reduces the risk of landslides.

*Table 4. Landslide risk reduction strategies in the study area*

| Existing management strategies | Frequency | Percentage |
|---|---|---|
| Agroforestry | 123 | 32% |
| Relocation from high-risk zones | 92 | 24% |
| Terracing | 84 | 22% |
| Storm water drainage system | 77 | 20% |
| Planting tree along rivers banks | 8 | 2% |
| Total | 384 | 100% |

Source: Field study, September- October 2022 (Note: Multiple responses were considered)

The residents of Kivu catchment of Rwanda gave their feedback on how landslides are managed. The management of landslides includes the integration of all intervention of activities and control measures on the side of government and its partners. The assessment of locals' perception on landslide management revealed

that 96% of respondents agreed that landslide control techniques are effective, indicating that residents have a positive opinion of their community's approach to managing landslides (Figure3). Additionally, it was shown that 63% of locals are happy with the landslide management methods that have been implemented in the research area, while 37% are not (Figure4).

Additionally, it was found that the government's involvement in landslide control is excellent while that of

NGOs is subpar or nonexistent in some areas of the catchment (Fig 5&6). It is clear that central government disaster management strategy is crafted using a methodology where all partners are involved in each and every intervention. The failure of all stakeholders to work together, however, prevents the policy from being successfully implemented at the ground level. The survey's results regarding the efficiency of landslide risk reduction measures, the satisfaction of locals with the implementation of these measures, and the involvement of

both NGOs and the local government revealed gaps in cooperation between these parties where NGOs do not appear to be active participants during intervention activities for landslide management.

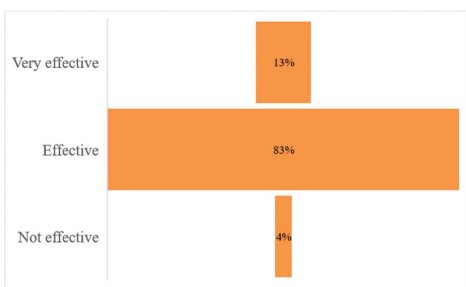

*Figure 3. Effectiveness of landslide control measures measures*

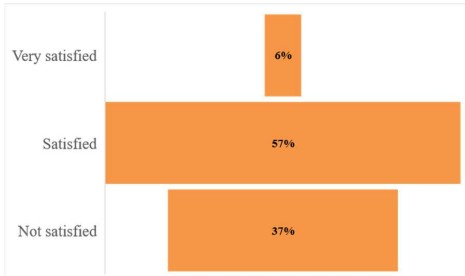

*Figure 4. Locals' satisfaction with landslides control*


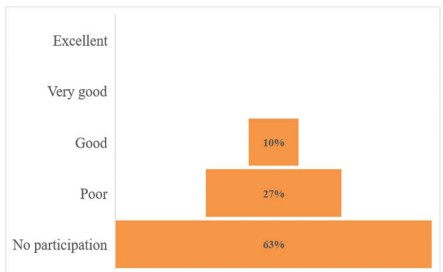

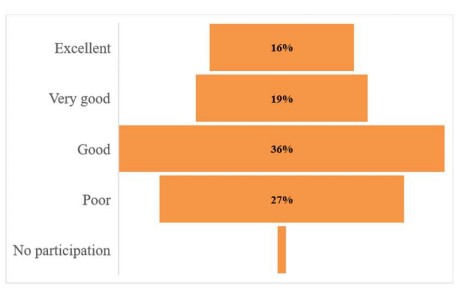

*Figure5. NGOs participation in landslides management*  *Figure 6. Government participation in landslides management*

The results of the descriptive method on the influence of landslide causative and triggering elements showed that heavy rain is the first in terms of its effectiveness in causing landslides to occur in Kivu catchment of

Rwanda. Heavy rain received the highest percentage ranking (96.1%), followed by steep slope (74.5%), roads construction (27.6%), improper agricultural practices (26%), deforestation (23.2%), climate change (6.8%), earthquake (4.9%), mining (3.4%), and unorganized settlement (1.3%), in that order.

*Table 5: Factors of landslide frequencies*

| | | Responses | | Percent of Cases |
|---|---|---|---|---|
| | | N | Percent | |
| | Heavy rain | 369 | 36.4% | 96.1% |
| | Inappropriate Agricultural practices | 100 | 9.9% | 26.0% |
| | agriculture | 2 | 0.2% | 0.5% |
| | Steep slope | 286 | 28.2% | 74.5% |
| Factors of Landsildes[a] | Road's Construction | 106 | 10.4% | 27.6% |
| | Deforestation | 89 | 8.8% | 23.2% |
| | Mining | 13 | 1.3% | 3.4% |
| | Climate change | 26 | 2.6% | 6.8% |
| | Earthquake | 19 | 1.9% | 4.9% |
| | Unorganized settlement | 5 | 0.5% | 1.3% |
| Total | | 1015 | 100.0% | 264.3% |
| a. Dichotomy group tabulated at value 1. | | | | |

Source: Field study, September- October 2022

The statistical approach was used to conduct a quantitative investigation of the association between causal factors and landslide occurrences in the study area. The interaction between the dependent variable (information on landslide risks) and the independent variables (heavy rain, improper agricultural methods, steep slopes, road

construction, deforestation, mining, climate change, earthquake, and unorganized settlements) demonstrates the influence of each variable on the others. A standardized Beta coefficient, obtained from the regression analysis done in this study, compares the magnitude of each independent variable's influence on the dependent variable; and the effect is bigger as the absolute magnitude of the Beta coefficient is higher. Based on the regression analysis on landslide occurrences and causal factors, and by considering significant factors and its 95%

confidence interval. It is clear that inappropriate agriculture practices, roads construction and deforestation have a highly significant effect on landslide occurrences with 95% confidence intervals of [-0.349, -0.120], [0.200, 0.408] and [-0.293, -0.061] respectively. The results also showed that climate change and earthquake have a




significant effect on landslide occurrences with 95% confidence intervals of [0.081,0.443] and [-0.520, -0.090] respectively. However, other factors such as heavy rainfall, steep slope, mining, and unorganized settlement did

not show a significant effect on landslide occurrences in the study area.

The primary cause factors for landslides in the Kivu catchment of Rwanda, according to a qualitative analysis of landslide frequency factors, were determined to be high rainfall and a steep slope. However, quantitative research showed that none of the two causes is significantly responsible for causing landslides in the study area. According to research findings on landslides in the western part of Rwanda that were based on prior literature,

heavy rain is one of the main causes of landslides (Bizimana, 2015; Nsengiyumva,2020).

By examining the socioeconomic factors of the local community that were taken into consideration for this study, as well as how residents of the research area explained landslide-associated factors from their own perspectives, it may be possible to address the discrepancy between the qualitative and quantitative results for this study and create a scientific discussion platform.

*Table 6. Regression analysis on landslide occurrences and causative factors*

| Coefficientsa | | | | | | | |
|---|---|---|---|---|---|---|---|
| Model | Unstandardized Coefficients | | Standardized Coefficients | t | Sig. | 95.0% Confidence Interval for B | |
| | B | Std. Error | Beta | | | Lower Bound | Upper Bound |
| (Constant) | .671 | .136 | | 4.936 | .000 | .403 | .938 |
| Heavy rain | -.112 | .128 | -.043 | -.874 | .382 | -.364 | .140 |
| Inappropriate Agricultural practices | -.234 | .058 | -.206 | -4.021 | .000 | -.349 | -.120 |
| agriculture | -.171 | .344 | -.025 | -.496 | .620 | -.847 | .505 |
| Steep slope | -.102 | .055 | -.089 | -1.851 | .065 | -.210 | .006 |
| Road's Construction | .304 | .053 | .272 | 5.736 | .000 | .200 | .408 |
| Deforestation | -.177 | .059 | -.150 | -2.996 | .003 | -.293 | -.061 |
| Mining | .003 | .128 | .001 | .024 | .981 | -.249 | .255 |
| Climate change | .262 | .092 | .132 | 2.844 | .005 | .081 | .443 |
| Earthquake | -.305 | .109 | -.133 | -2.795 | .005 | -.520 | -.090 |
| Unorganized settlement | -.153 | .207 | -.035 | -.743 | .458 | -.560 | .253 |

a. Dependent Variable: Info on landslide occurrences/ risk

Source: Field study, September- October 2022

Analytically, it is reasonable to assume that respondents' educational background (level and field) may have an impact on how truthfully, they answered the survey questions and the level of awareness that underlay those

answers throughout the entire survey process. The task a researcher does while acquiring information is another area to look at. Here, the interaction between a researcher and the local community that was interviewed for this study may have some task-giving potential in terms of determining the true meaning of the respondents' responses from their own appellation to the true sense of what could be considered by a researcher through his knowledge and expertise in this study. Thus, contrary to what the local community confirmed, the main causes

of landslides in Kivu catchment of Rwanda are not heavy rain and steep slope. In contrast, statistical study has shown that deforestation, inappropriate agricultural practices, roads construction, earthquakes, and climate





change are the key factors contributing to the occurrence of landslides in Kivu catchment of Rwanda. In order to improve the literature already available on landslides in the Kivu catchment of Rwanda, further research must be conducted on the subject utilizing a range of approaches, especially those that are related to artificial intelligence and its usage for analyzing landslides in the study region.

### 4. Conclusion

The perception of landslides in the local community can be analyzed from a variety of perspectives, including socioeconomic, cultural, political, and others. For these reasons, research should be thoroughly based on the viewpoint of the local community as well as a multidimensional analysis of the researcher that integrates all relevant factors to produce accurate results. In Kivu catchment of Rwanda, landslide risk reduction and adaptation strategies exist and their implementation by the local government and its partners is efficient and satisfactory according to the local community's appreciations. However, there are still gaps in cooperation between these parties where Non-Governmental Organizations do not appear to be active participants during intervention activities for landslide management in the Kivu catchment of Rwanda.

**Acknowledgments:** This work was supported by The Regional Scholarship and Innovation Fund -Partnership for Skills in Applied Sciences, Engineering and Technology (RSIF-PASET).

**Author contributions**: All authors conceived and wrote the manuscript text.

**Conflict of interest**: The authors declare no competing interests.

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
