# Peer review of "Local perception and adaptation strategies to landslide occurrence in the Kivu catchment of Rwanda"

_Natural Hazards and Earth System Sciences, 2023_

## Referee Comment (RC3)

**Manuscript Number:**

nhess-2023-47

**Title:**

Local perception and adaptation strategies to landslide occurrence in the Kivu catchment of Rwanda

**Overview and general comments:**

Referring to the interviews, the authors evaluate the local perception and adaptation strategies to landslides along Kivu Lake in Rwanda. They provided valuable information regarding how locals see the landslide threat and what should be done. As suggested by Gill et al. (2021)*, "understanding and listening to stakeholders" is a crucial aspect of addressing natural hazard threats adequately and building sustainable and resilient communities; the study makes an important contribution not only relevant to their research area but also for the global landslide risk community.

In places, the tone of the manuscript misinforms the reader through people's perceptions to sound like reality. Authors should pay attention to keeping the perceptions and facts separate. Below I provide all my major comments that need to be addressed to elevate the quality of the research work to the NHESS standards. I also provided several minor concerns in my attached document. Please refer to them for my line-by-line suggestions.

- The introduction inefficiently leads to research questions and the manuscript's goals. Maybe more than half of the information is irrelevant to the study goals. I recommend trimming this section considerably.
- A major relevance of this study is the collected data (e.g., questionaries and interviews); hence I strongly encourage the authors to provide the collected data as supplementary information. Providing the data will not only increase the credibility of the current work but also will attract more attention to the manuscript.
- At the end of the results section, section authors highlight the main landslide controlling factors from the locals' perspective. They should extend this section and compare the perceptions to reality (referring to literature, I provided several options in my detailed comments) in a dedicated discussion section. I believe this is one of the most exciting outcomes of their study.
- In a dedicated discussion section, authors should also highlight the relevance of their observations/results in the broader context. What new knowledge have they provided for the global landslide community? What are the main limitations of the landslide research in the study area? And what should be done next?
- Authors should consider repeating their statistical assessment for different education groups (or considering other relevant divisions such as age) to show a contrast between them, if there is any. They should also link these perceptions to reality in the dedicated discussion section.

* Gill, J. C., Taylor, F. E., Duncan, M. J., Mohadjer, S., Budimir, M., Mdala, H., and Bukachi, V.: Invited perspectives: Building sustainable and resilient communities – recommended actions for natural hazard scientists, Nat. Hazards Earth Syst. Sci., 21, 187–202, https://doi.org/10.5194/nhess-21-187-2021, 2021.

**Detail Comments**

**Abstract:**

The abstract should start with a motivational sentence highlighting the topic's relevance.

Line 10: What are the qualitative, quantitative, and combined methods? Could you please provide them all or at least some examples?

Line 11: Keyword "Origin" → do the authors here mean triggers, causes, or both (controlling factors)?

Lines 14 and 15: Heavy rain and earthquake are not landslide causes. They are landslide triggers.

Line 17: Providing adopted behaviour examples, consider providing the most important ones with numerical data.

The provided results are primarily generic and uninformative. Please consider giving key results with numerical supporting information. For example, sentences like the following would be beneficial "XYZ% of the people perceive that heavy rainfall is the most important factor controlling landslides accurately. Whereas XYZ% thinks falsely that Earthquakes and Deforestation are the key factors. Among the most educated people, the perception XYZ% better represents the reality." These are just some examples; please consider them cautiously.

The last sentence of the abstract should provide the potential use of the results instead of further research suggestions, which better fit the sections conclusions or discussions.

**Introduction:**

The introduction inefficiently leads to research questions and the manuscript's goals. Maybe more than half of the information is irrelevant to the study goals. I recommend trimming this section considerably.

The paragraphs between 30 and 48 are somewhat irrelevant to the current study.

Is the paragraph's information between lines 55 to 60 relevant to the current study?

The majority of the information in the paragraph between lines 66 to 77 provided information about the landslide death toll in regions that are neither relevant to the study area nor the manuscript's subject. For example, why do authors mention the global death toll in 2016 and from 2004 to 2010, and why do they list fatalities from the US and Sri Lanka? Why the information regarding 3 billion people being impacted by disasters is relevant? I guess the study does not focus on earthquake or tsunami impact. I suggest removing these statements entirely

Line 75: the statement needs a reference "landslide have grown…". For example,

Cendrero, A., Forte, L. M., Remondo, J., and Cuesta-Albertos, J. A.: Anthropocene Geomorphic Change. Climate or Human Activities?, Earth's Future, 8, https://doi.org/10.1029/2019EF001305, 2020.

Line: 82: "…could minimize…" Is it "minimiz**ed**"? Or better "mitigated"? I believe the source article refers to achieved aspects.

Line 116: "…the inventory is difficult." In which aspect is it difficult?

The introduction does not refer to crucial references relevant to the manuscript's core topic. Please consider supporting your claims with the below suggestions and more. These could also help shaping your discussion section. If you don't have access to the suggested articles below, please do not hesitate to approach me.

Bozzolan, E., Holcombe, E., Pianosi, F., and Wagener, T.: Including informal housing in slope stability analysis–an application to a data-scarce location in the humid tropics, NHESS, 20, 3161–3177, https://doi.org/10.5194/nhess-20-3161-2020, 2020.

Bozzolan, E., Holcombe, E. A., Pianosi, F., Marchesini, I., Alvioli, M., and Wagener, T.: A mechanistic approach to include climate change and unplanned urban sprawl in landslide susceptibility maps, Science of The Total Environment, 858, 159412, https://doi.org/10.1016/j.scitotenv.2022.159412, 2023.

Smith, H., Coupé, F., Garcia-Ferrari, S., Rivera, H., and Castro Mera, W. E.: Toward negotiated mitigation of landslide risks in informal settlements: reflections from a pilot experience in Medellin, Colombia, E&S, 25, art19, https://doi.org/10.5751/ES-11337-250119, 2020.

Depicker, A., Jacobs, L., Mboga, N., Smets, B., Van Rompaey, A., Lennert, M., Wolff, E., Kervyn, F., Michellier, C., Dewitte, O., and Govers, G.: Historical dynamics of landslide risk from population and forest-cover changes in the Kivu Rift, Nat Sustain, 4, 965–974, https://doi.org/10.1038/s41893-021-00757-9, 2021.

Dewitte, O., Dille, A., Depicker, A., Kubwimana, D., Maki Mateso, J.-C., Mugaruka Bibentyo, T., Uwihirwe, J., and Monsieurs, E.: Constraining landslide timing in a data-scarce context: from recent to very old processes in the tropical environment of the North Tanganyika-Kivu Rift region, Landslides, 18, 161–177, https://doi.org/10.1007/s10346-020-01452-0, 2021.

Dille, A., Dewitte, O., Handwerger, A. L., d'Oreye, N., Derauw, D., Ganza Bamulezi, G., Ilombe Mawe, G., Michellier, C., Moeyersons, J., Monsieurs, E., Mugaruka Bibentyo, T., Samsonov, S., Smets, B., Kervyn, M., and Kervyn, F.: Acceleration of a large deep-seated tropical landslide due to urbanization feedbacks, Nat. Geosci., https://doi.org/10.1038/s41561-022-01073-3, 2022.

Froude, M. J. and Petley, D. N.: Global fatal landslide occurrence from 2004 to 2016, Nat. Hazards Earth Syst. Sci., 18, 2161–2181, https://doi.org/10.5194/nhess-18-2161-2018, 2018.

Gill, J. C., Taylor, F. E., Duncan, M. J., Mohadjer, S., Budimir, M., Mdala, H., and Bukachi, V.: Invited perspectives: Building sustainable and resilient

communities – recommended actions for natural hazard scientists, Nat. Hazards Earth Syst. Sci., 21, 187–202, https://doi.org/10.5194/nhess-21-187-2021, 2021.

Mateos, R. M., Herrera, G., García-Davalillo, J. C., Grandjean, G., Poyiadji, E., et al.: Integration of Geohazards into Urban and Land-Use Planning. Towards a Landslide Directive. The EuroGeoSurveys Questionnaire, in: Advancing Culture of Living with Landslides, edited by: Mikos, M., Tiwari, B., Yin, Y., and Sassa, K., Springer International Publishing, Cham, 1067–1072, https://doi.org/10.1007/978-3-319-53498-5_121, 2017.

Nsabimana, J., Henry, S., Ndayisenga, A., Kubwimana, D., Dewitte, O., Kervyn de Meerendré, F., and Michellier, C.: Exposure to past disasters related to hydrological hazards: the case of Bujumbura city, Burundi, display, https://doi.org/10.5194/egusphere-egu22-8091, 2022.

Ozturk, U.: Geohazards explained 10: Time-dependent landslide susceptibility, Geology Today, 38, 117–120, https://doi.org/10.1111/gto.12391, 2022.

Ozturk, U., Bozzolan, E., Holcombe, E. A., Shukla, R., Pianosi, F., and Wagener, T.: How climate change and unplanned urban sprawl bring more landslides, Nature, 608, 262–265, https://doi.org/10.1038/d41586-022-02141-9, 2022.

Raška, P., Riezner, J., Bíl, M., and Klimeš, J.: Long-term landslide impacts and adaptive responses in rural communities: Using historical cases to validate the cumulative causation approach, International Journal of Disaster Risk Reduction, 93, 103748, https://doi.org/10.1016/j.ijdrr.2023.103748, 2023.

**Data collection:**

Line 134: A sample size of 384 sounds somewhat too small to represent >2M people. Could the authors elaborate on this choice?

Line 136: "…different cells." What are those cells referred to?

Lines 150 and 151: A major relevance of this study is the collected data. Hence, I strongly encourage the authors to provide the collected data as supplementary information. Providing the data behind the study will not only increase the credibility of the current work but also will attract more attention to the manuscript.

Line 152: "…by the researcher." Who is the researcher?

Line 153: "…research assistants…". Since the current study would not be possible without the contribution of these assistants, it might be an option to name and list them in the acknowledgements or the supplementary material (considering the number of people involved).

Line 160: I appreciate that the authors mention ethical aspects of the study. Please consider referring to the

Gill, J. C., Taylor, F. E., Duncan, M. J., Mohadjer, S., Budimir, M., Mdala, H., and Bukachi, V.: Invited perspectives: Building sustainable and resilient communities – recommended actions for natural hazard scientists, Nat. Hazards Earth Syst. Sci., 21, 187–202, https://doi.org/10.5194/nhess-21-187-2021, 2021.

**Results and Discussion:**

To distinguish the contributions of the current study from the interpretations and limitations, please separate the results and discussion sections.

Lines 165 and 176: Please use active sentences to ensure the reader that these results are your contributions. For example, in line 172, "it was found that…" this part of the sentence is unnecessary. Also, the information provided here resembles Table 1; hence please consider trimming it and providing only the key aspects here, while the rest is in Table 1 already.

Line 182: heavy rain is not a landslide cause; it is a landslide trigger, so as the earthquakes. An alternative word could be landslide-relevant "factors". Please use the terms correctly and consistently throughout the manuscript.

Line 185: "…among hilly…" Besides rugged terrain, the most critical landslide control is the hillslope angle. Could you please consider providing the hillslope angle distribution in a figure?

Line 203: "…an active volcano of Nyiragongo may be justified to be exposed to landslides induced by earthquake." Although I am not an expert on the terminology on these subjects, I believe the authors do not refer to Earthquakes but Volcanic-activity induced seismicity (In a figure where elevation or slope is shown, this volcano could be highlighted.).

Since the authors collected landslide data from the field, they should provide the landslide data in a shape file as a supplement along with the manuscript. It would be nice to have a figure showing their location (or estimated location) on a map. In the background elevation or another important parameter could be presented, e.g., elevation, slope, or mean annual precipitation?

Lines between 229 and 235: The information might better fit the introduction.

Lines between 250 and 253: could you please explain why the community is satisfied and why not for all the aspects mentioned?

A figure should support results provided between lines 277 and 290 (please transform Table 6 to a figure.). The statements in this paragraph are absolute. Authors should highlight the fact that it is how locals observe landslide risk. Their perception might not represent reality. In a dedicated discussion section, authors should consider further referring to literature to compare people's perceptions to the fact. Accordingly, they should thoroughly discuss why the perception and the reality are different or similar.

Line 293: "…two causes…" What are they?

Lines between 303 and 315: Could the authors repeat the statistical assessment for different education groups (or consider other relevant divisions such as age)? Repeating these analyses will aid in seeing the contrast in-between different categories if there is any. Linking these various perceptions to reality in the dedicated discussion section would immensely increase the current work's contribution to global landslide research.

**Conclusion:**

The concluding remarks highlighting research gaps sound like suggestions that better suit the discussion section. Here authors should better highlight the contributions of their work. It would be nice if they could support these outcomes with their observed numerical data.

**Tables:**

In general, tables discourage readers from reviewing the provided information. Hence, I strongly recommend converting tables to figures if possible.

Table 1: The information provided in this table is essential to interpret the results of the current study. However, they could have been presented as more reader-friendly in a figure. Please consider transforming it.

Tables 2, 3, 4, 5, and 6 could also be transformed into figures.

**Figures:**

Figure 1: Labels are missing, ticks, legend, and the text in the figure are small to read. Legend might be redundant since the district names are written on the map already.

Figure 2: This figure is rather uninformative; it could be omitted.

Figures 3 and 4 can be combined.

Figures 5 and 6 can be combined.